# Production of Cellulosic Ethanol from Enzymatically Hydrolysed Wheat Straws

**Vasile-Florin Ursachi and Gheorghe Gutt ***

Faculty of Food Engineering, Stefan cel Mare University of Suceava, 720229 Suceava, Romania; florin.ursachi@fia.usv.ro

**\*** Correspondence: g.gutt@fia.usv.ro

**Abstract:** The aim of this study is to find the optimal pretreatment conditions and hydrolysis in order to obtain a high yield of bioethanol from wheat straw. The pretreatments were performed with different concentrations of sulphuric acid 1, 2 and 3% (*v/v*), and were followed by an enzymatic hydrolysis that was performed by varying the solid-to-liquid ratio (1/20, 1/25 and 1/30 g/mL) and the enzyme dose (30/30 μL/g, 60/60 μL/g and 90/90 μL/g Viscozyme® L/Celluclast® 1.5 L). This mix of enzymes was used for the first time in the hydrolysis process of wheat straws which was previously pretreated with dilute sulfuric acid. Scanning electron microscopy indicated significant differences in the structural composition of the samples because of the pretreatment with $H_2SO_4$ at different concentrations, and ATR-FTIR analysis highlighted the changes in the chemical composition in the pretreated wheat straw as compared to the untreated one. HPLC-RID was used to identify and quantify the carbohydrates content resulted from enzymatic hydrolysis to evaluate the potential of using wheat straws as a raw material for production of cellulosic ethanol in Romania. The highest degradation of lignocellulosic material was obtained in the case of pretreatment with 3% $H_2SO_4$ (*v/v*), a solid-to-liquid ratio of 1/30 and an enzyme dose of 90/90 μL/g. Simultaneous saccharification and fermentation were performed using *Saccharomyces cerevisiae* yeast, and for monitoring the fermentation process a BlueSens equipment was used provided with ethanol, $O_2$ and $CO_2$ cap sensors mounted on the fermentation flasks. The highest concentration of bioethanol was obtained after 48 h of fermentation and it reached 1.20% (*v/v*).

**Keywords:** wheat straws; pretreatment; hydrolysis; fermentation; bioethanol

## 1. Introduction

In the past decade, due to climate changes there has been an increasing attention on reducing greenhouse gas (GHG) emissions. In accordance with the Paris Agreement (Council Decision (EU) 2016/1841 of the 5 October 2016) from 2023, every 5 years a comprehensive assessment of the progress of the parties will be made on the basis of scientific data and the situation regarding the reduction of emissions, the adjustments made and the support provided will be analysed. Compared to 1990, the mandatory target set for 2030 is at least a 40% domestic reduction economy-wide greenhouse gas (GHG) emissions [1].

Agricultural biomass is considered one of the most important renewable energy resources and contributes to the development of bioenergy generation. It consists of annual and perennial energy crops (green biomass for animals feed), residues from agricultural production (straw, corn stalks, corn cobs, sugar cane, etc.) and the food industry (residues from dairy industry, sugar industry, etc.). In recent years, there has been an increasing trend in obtaining bioethanol from renewable resources such as straws resulted from cereals harvesting. However, huge quantities of wheat straws (WS) are generated annually which could be used for the production of cellulosic bioethanol [2,3].

Liquid biofuels can partially or completely replace conventional fuels and can be an alternative source in the transport sector (aviation, shipping and heavy freight trucks). Therefore, liquid biofuels are an important solution because they do not require major changes in distribution infrastructure or the transport fleet. International Renewable Energy Agency (IRENA) argues that a reduction in carbon emissions by 2050 is only possible if there is a five-fold increase in biofuel consumption, from 130 billion litres in 2016 to almost 650 billion litres in 2050 [4].

Wheat (Triticum sp.) is the third most cultivated cereal in the world. Based on the data provided by Food and Agriculture Organization of the United Nations (FAOSTAT) in 2020, the worldwide cereal production is 2789.8 million tons. The worldwide cereal production in 2020 (2789.8 million tons) is higher by 3% compared to 2019 (2708.5 million tons) [5]. In 2019/2020 the estimated global wheat production is 762.2–764.1 million tons in comparison with the production of 2018/2019 which was 732.1 million tons (Figure 1) [6,7]. Therefore, if we take into account a coefficient of 1.3 [8,9], in 2019/2020 an amount of 990.86–993.33 million tons of wheat residues were produced.

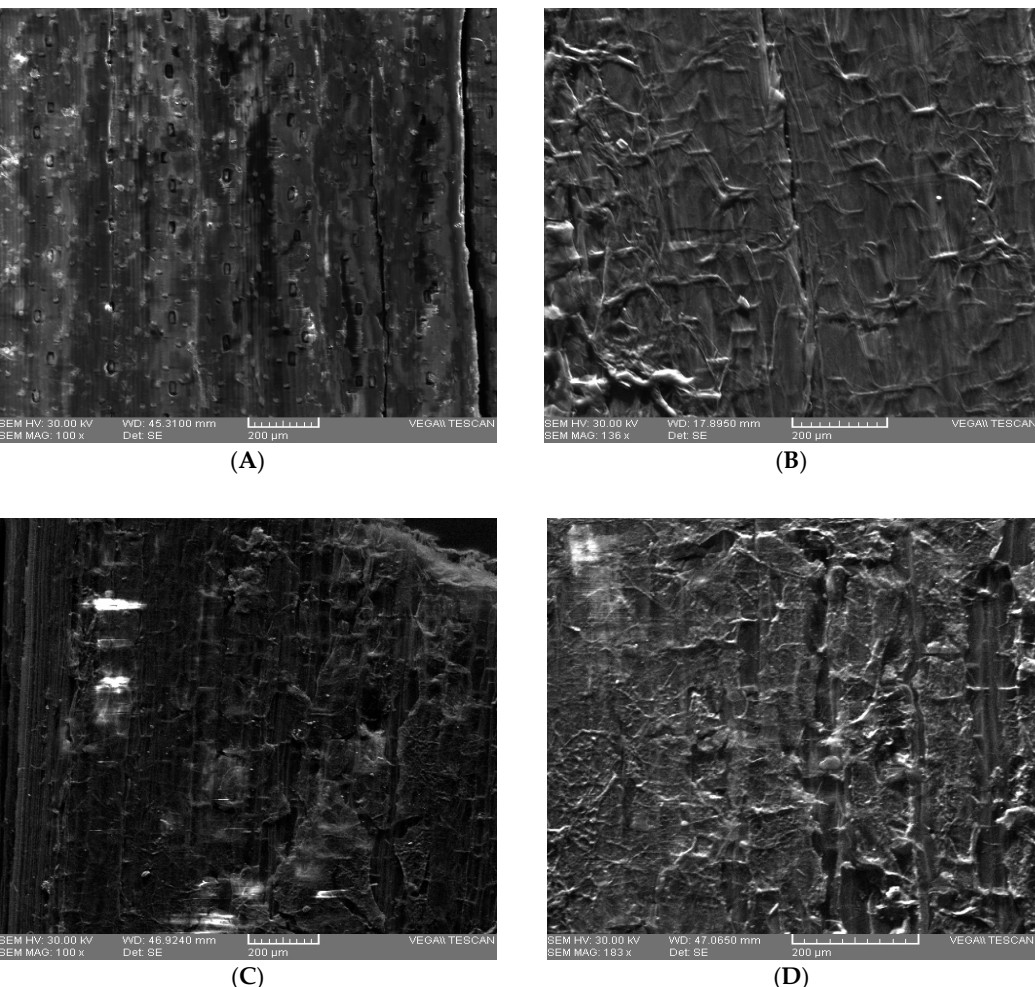

**Figure 1.** SEM images for untreated WS ((**A**)—E detector-exterior view; and for pretreated WS ((**B**) —SE detector-1% $H_2SO_4$ (*v/v*); (**C**)—SE detector-2% $H_2SO_4$ (*v/v*); (**D**)—SE detector–3%; $H_2SO_4$ (*v/v*)).

The wheat straw is consisted of internodes (57 ± 10%), nodes (10 ± 2%), leaves (18 ± 3%), straw (9 ± 4%) and the central axis (6 ± 2 %). The chemical composition of the wheat straw contains cellulose (34–40%), hemicellulose (20–25%) and lignin (20%) [10–13].

Cellulose is a linear polymer composed of β-*D*-glucopyranosyl units linked at 1,4 positions [14]. The utilisation of cellulose and hemicellulose sugars present in the hydrolysate of lignocellulosic biomass is essential for the production of bioethanol [15]. Hemicellulose is a complex polysaccharide, found in

the cell walls of plants, composed of neutral monosaccharide units: *D*-xylose, *D*-manose, *D*-galactose, *D*-glucose, *L*-arabinose, *D*-glucuronic acid etc. Hemicellulose fractions consist of β-*D*-xylopyranose backbone residues linked at 1,4 positions to the units of *L*-arabinose, *D*-galactose and/or *D*-glucuronic acid [16].

The mechanical pretreatment of lignocellulosic materials is an important stage in the process of obtaining bioethanol which reduces the particle size and crystallinity of cellulose, increases the contact surface, dissociates tissues and disintegration of the cell wall [17–19]. All these parameters contribute significantly to the conversion of saccharides during hydrolysis. Talebnia et al. (2010) reported in their study that mechanical pretreatment (chopping and grinding) contributes significantly to the improvement of enzymatic hydrolysis of WS. It was established that, after 24 h of hydrolysis, the content of glucose and xylose was increased by 39% and 20% respectively when the particle size was reduced from 2–4 cm to 53–149 μm [8]. However, the very small particles is a disadvantage because it leads to a high energy consumption in the milling phase. On the other hand, the additional elimination of lignin contributes significantly to the decrease of energy consumption [20].

The acid pretreatment leads to the formation of inhibitory compounds at a low value of pH, high temperature and pressure. The degradation of lignin and hemicellulose produces free organic acids (acetic, formic and levulinic acid) or phenolic derivatives (4-hydroxynenzoic acid or vanillin), and 2-furaldehyde (furfural–FF) with 5-hydroxymethyl-2-furaldehyde (hydroxymethylfurfural–HMF) which affect negatively the chemical composition, enzymatic hydrolysis and fermentation. The FF and HMF are obtained from the transformation of pentoses (xylose and arabinose) and hexoses (glucose, mannose and galactose) [21]. Acid pretreatment is a significant phase for obtaining bioethanol which destroys the crystalline structures of cellulose and increases the accessibility of the enzyme during enzymatic hydrolysis [22,23]. Zheng et al. (2017) studied the impact of acid pretreatment on WS, using 2% and 4% of $H_2SO_4$. This pretreatment led to the elimination of high amount of hemicellulose [24]. The acid pretreatment is most suitable for WS, but also has some disadvantages, such as acid pollution of the environment and feed, and production of secondary compounds [25–27]. Tian et al. (2018) established that pretreatment with 2% diluted $H_2SO_4$ improved the rate of lignin removal from WS [27]. Mardetko et al. (2018) reported that pretreatment with 0.5% $H_2SO_4$ was more effective in obtaining a higher amount of glucose (4.84 g/L) for 10 min at 200 °C compared to 4.09 g/L glucose obtained by using 2% $H_3PO_4$ [28].

During pretreatment, HMF can be converted to formic and levulinic acid, while FF to formic acid [19]. The structure of hemicellulose is composed of acetylated sugars which can turn into acetic acid during pretreatment. Also, during lignin pretreatment, significant amount of phenolic and aromatic compounds was achieved [29]. These compounds limit the transformation of sugars into bioethanol which reduce the final yield of this alcohol.

Enzymatic hydrolysis is the most efficient method of releasing carbohydrates from lignocellulosic materials. Thus, the hydrolysis of cellulose is catalysed by a class of enzymes, called cellulases. This type of hydrolysis is influenced by the lignocellulosic substrate, enzyme activity and process conditions [8]. By hydrolysis, high concentrations of acetic and formic acids were detected, with values between 4.9–9.4 g/L and 1.6–10.3 g/L, respectively, and levulinic acid of 0.3–0.6 g/L. Acetic acid, furfural and HMF are the products that result from hydrolysis and has inhibitory effects on the fermentation process [30].

The aim of this paper is to obtain bioethanol as a result of the superior recovery of wheat straw. A number of objectives have also been set for monitoring the whole process, such as the pretreatment, enzymatic hydrolysis and fermentation steps. To find out which is the most indicated option for pretreatment of wheat straws and its effect on the hydrolysis step, a diluted concentration of 1, 2 and 3% (*v/v*) $H_2SO_4$ was used.

## 2. Materials and Methods

*2.1. Materials*

Wheat straws (WS) used in this study were collected on 21st of July 2020, from Mitocu Dragomirnei (Suceava county, Romania, 47°45′14.0″ N 26°13′44.0″ E).

The analytical reagents, standard materials and enzymes used in this study were purchased from Sigma-Aldrich. The utilised enzymes Viscozyme® L and Celluclast® 1.5 L (Novozyme Corp, Bagsvaerd, Denmark) were used for enzymatic hydrolysis. The yeast, fermentation activator and diammonium phosphate (DAP) were purchased from Enzymes & Derivates, Neamț. Milli-Q water (Direct-Q® 3 UV, Milipore SAS 67120, Molsheim, France) was used in the preparation of reagents, standards and samples.

Viscozyme® L contains a multienzymatic complex of carbohydrates such as arabanase, cellulase, β-glucanase, hemicellulase and xylanase. This is a clear liquid enzyme produced by *Aspergillus aculeatus* which has brown color and has a density of approx. 1.2 g/mL, enzymatic activity ≥ 100 FBG/g (β-glucanase fungal units) with activity under optimal conditions at pH between 3.3 and 5.5 and a temperature of 25–55 °C. The enzyme must be stored at a temperature of 2–8 °C.

Celluclast® 1.5 L is an enzyme (endoglucanase) that hydrolyses the (1,4)-β-D-glucoside bonds in cellulose and other β-glucans. This is a brown liquid enzyme and is produced by *Trichoderma reesei*, has a density of approx. 1.22 g/mL, enzymatic activity ≥ 700 EGU/g (β-glucanase fungal units) with activity in optimal conditions at pH between 4 and 6 and a temperature of 25–55 °C. At lower temperatures (5–10 °C) the shelf life is considerably increased.

Hydra PC is a yeast activator that helps strengthen its plasma membrane and gives it increased resistance in unfavourable environments. This product contains a significant amount of magnesium that contributes to cell division, increases the speed of yeast development.

DistillaMax SR is a special yeast of the species *Saccharomyces cerevisiae* that produces low levels of higher alcohols and has good resistance to osmotic pressure, organic acids and high temperatures. The recommended amount is between 10 and 50 g/hL, and the fermentation temperature is 30–35 °C.

*2.2. Methods*

2.2.1. Chemical Composition of WS

Total of 5 g of ground WS was accurately weighed on analytical balance (Partner Corporation, Bracka 28, Poland) and heated at 105 °C for 4 h to a constant mass in an oven [31]. Then, the ash content was determined by weighing 1 g of ground WS and calcining at 575 ± 25 °C for 3 h [32] in a furnace (Thermo Scientific Thermolyne, Kerper blvd Dubuque, Iowa, USA). The content of cellulose [31] and lignin [32] was determined using the methods described by Ishtiaq et al. (2010) and Sluiter et al. (2012) respectively.

2.2.2. Pretreatments of WS

A. Mechanical pretreatment. The WS were cut into small pieces and then heated at 40 °C for 24 h in an oven (Memmert, Schwabach, Germany). Then, WS were milled and sieved in a shaker (Sieve shaker Retsch, Haan, Germany). In this study, the particle size used of wheat straws grounded (WSG) was <1 mm.

B. Physico-chemical pretreatment. Total of 2 g of WSG were weighed with accuracy and precision of 0.001. The samples were added in borosilicate glass bottles with polypropylene cap and pouring ring and were boiled for 1 h at 100 °C in a water bath (Precisdig JP Selecta, Abrera, Barcelona, Spain) with different concentration of $H_2SO_4$ (1, 2 and 3% (*v/v*)) and solid/liquid (S/L) ratio (1/20, 1/25 and 1/30 *w/v*). Then, the samples were rapidly cooled using cold water.

### 2.2.3. Enzymatic Hydrolysis of Pretreated WS

The preatreated samples were filtered under vacuum using Whatman qualitative filter paper, grade 5 and washed with 100 mL of Milli-Q water to remove enzymatic inhibitory compounds. Then, the solid fraction was transferred into borosilicate glass bottles with polypropylene cap and pouring ring, and Milli-Q water was added over the solid fraction, maintaining the S/L ratio of 1/20, 1/25 and 1/30 respectively. The pH of obtained samples was corrected to 4.5 with 3 M NaOH using a pH meter (Mettler-Toledo, model SevenCompact S210). The enzyme activities were described by Ghose (1987) and used by Vintilă et al. (2019) [33–35]. Afterwards, 30/30 µL/g or 20 FPU/g, 60/60 µL/g or 40 FPU/g and 90/90 µL/g or 60 FPU/g (Viscozyme® L and Celluclast® 1.5 L) were dosed in the pretreated WS. The order of addition of the enzymes was Viscozyme® L and Celluclast® 1.5 L, respectively, which were allowed to act for 24 h at 52 °C.

### 2.2.4. Scanning Electron Microscope (SEM) Analysis

The acid pretreated and enzyme-hydrolysed samples were dried and analysed with SEM Tescan Vega II LMU (Tescan Orsay Holding, Brno, Czech Republic), operated at 30 kV.

### 2.2.5. ATR-FTIR Analysis

FT-IR analysis was performed to detect the modifications in the functional groups in dried raw material, pretreated with acid and enzyme-hydrolysed WS. The FT-IR spectra of samples were achieved with a FT-IR spectrometer (Thermo Scientific, Karlsruhe, Dieselstraße, Germany) with ATR IX option. The results were obtained within a range of 400–4000 cm$^{-1}$ with a detector at 4 cm$^{-1}$.

### 2.2.6. HPLC Instrumentation and Separation Conditions

The individual phenolic compounds, organic acids and individual carbohydrates were analysed using a high-performance liquid chromatography (HPLC) (Shimadzu, Kyoto, Japan) equipped with a LC-20 AD liquid chromatograph, SIL-20A autosampler, CTO-20AC coupled with a SPD-M20A diode array detector (DAD) and RID-10A refractive index detector (RID) respectively. The separation of phenolic compounds, organic acids and individual carbohydrates was performed in a column specific to each constituent analysed. The standards of phenolic compounds, organic acids and individual carbohydrates were determined based on the retention times and quantified based on their calibration curves (all the curves had R$^2$ higher than 0.98). For samples, the limits of detection (LOD) and limits of quantification (LOQ) were calculated according to Kuppusamy et al. (2018) [36–38].

Determination of Individual Phenolic Compounds

For the analysis of the individual phenolic compounds resulting from the acid pretreatment and from the enzymatic hydrolysates, the following was performed: 1 mL of solution from each sample pretreated with acid and respectively 1 mL of solution from the enzyme-hydrolysed samples for 24 h were filtered through PTFE membrane with 0.45-µm dimension of pores and were stored at −20 °C until analysis. The obtained samples were analysed using HPLC-DAD. The separation of individual polyhenols was performed into a Phenomenex Kinetex® 2.6 µm Biphenyl 100 Å column, LC Column 150 × 4.6 mm and thermostated at 25 °C (Column oven). The utilised method of analysis was described by Palacios et al. (2011) and used by Pauliuc et al. (2020) with some modifications [39–42]. The identification of the 12 phenolic compounds from pretreated and enzyme-hydrolysed samples of WS was performed at 280 nm for gallic, protocatechuic, vanillic and p-hydroxybenzoic acid; at 320 nm for chlorogenic, caffeic, p-coumaric and rosmarinic acid, quercitin, luteolin and kaempferol. The 12 phenolic compounds were injected individually to identify their retention time, then a mix of them was made. The final concentration of each identified phenolic compound was expressed in mg/L.

Determination of Organic Acids

In order to identify the organic acids resulting from the acid pretreatment and from the enzymatic hydrolysates, the following was performed: 1 mL of solution pretreated with acid samples and 1 mL of solution enzyme-hydrolysed samples for 24 h were filtered through PTFE membrane with 0.45 μm dimension of pores and were stored at −20 °C until analysis. The obtained mixtures were analysed using HPLC-DAD equipment. The separation of organic acids was performed into a Phenomenex Kinetex® 5 μm C18 100 Å HPLC Column 250 × 4.6 mm. The utilised method of analysis was described by Özcelik et al. [42,43]. The standards of individual organic acid (gluconic, acetic, formic, succinic, propionic, lactic and butyric acid) and mixed solution of them were prepared to determine the concentration of organic acids, expressed in mg/L.

Determination of Individual Carbohydrates After Enzymatic Hydrolysis

After 24 h of enzymatic hydrolysis, 1 mL of each sample was filtered through PTFE 0.45 μm dimension of pores and analysed for determination of carbohydrates using HPLC-RID. The separation of carbohydrates was performed in a Phenomenex Luna® Omega 3 μm de SUGAR 100 Å column, 150 × 4.6 mm. The utilised method of analysis was described by Bogdanov et al. (2002) [44]. Before analysis, the inactivation of enzymes was achieved by exposing the samples for 5 min at 121 °C [45]. The identification of carbohydrates was performed based on the standards (Carbohydrates Kit (CAR10-KIT) de D-(−)-Arabinose, ≥98% (A3131-5G), D-(−)-Ribose, ≥99%, (R7500-5G), D-(+)-Xylose, ≥99% (X1500-5G), D-(−)-Fructose, ≥99% (F0127-5G), D-(+)-Glucose, ≥99.5%, (G8270-5G-KC), D-(+)-Galactose ≥99.5% (G0750-5G), α-Lactose monohydrate, ≥99% total lactose basis, (L3625-5G), Sucrose ≥99.5% (S9378-5G-KC), D-(+)-Mannose, wood, ≥99% (M2069-5G), D-(+)-Maltose monohydrate, from potato, ≥99%, (M5885-5G) which were injected to identify the retention times, then a mixed solution of them was prepared to determinate the concentration of carbohydrates, expressed in mg/L.

2.2.7. Monitoring of Bioethanol Concentration

About 20 ± 0.001 g of WSG with <1 mm particle size was pretreated with 3% $H_2SO_4$ (*v/v*) following the procedure in Section 2.2.1, then the resulting solid fraction was hydrolysed using enzymes as described in Section 2.2.2. After 24 h of hydrolysis, the mash was cooled to 35 °C (optimal condition for yeast *Saccharomyces cerevisiae*). The pH of the mash was adjusted to 4.5 using a 3 M NaOH solution. The 2.5 g of Hydra PC activator was dissolved in 50 mL Milli-Q water (42 °C, conc. 5%), then the obtained mix was cooled to 38 °C and pH was adjusted to 4.5 using a solution of 3 M NaOH. Afterward, the cooled mix was used for the activation of 40 mg of dry yeast *Saccharomyces cerevisiae*. After 20 min, mash and 50 mL of solution with activated yeast were transferred to the fermentation flask. The temperature of the mash was maintained at 35 °C throughout the fermentation process.

The fermentation process was conducted using BlueSens gas sensors GmbH, Germany which monitors the content of carbon dioxide ($CO_2$), oxygen ($O_2$) and ethanol ($C_2H_5OH$) of the mash from lignocellulosic materials. The $CO_2$ (H31953 series), $O_2$ (H32132 series) and $C_2H_5OH$ (H32132) sensors were connected via the BACCom 12 multiplexer data which allows a connection of 12 sensors to the software. The processing and transmission data were performed in real time via BacVis software. The equipment used to monitor the fermentation process processes the information by means of three spectral sensors in the IR range, which are mounted on each hole of the fermentation vessel. BlueSens sensors allow continuous monitoring of the content of $CO_2$, oxygen and ethanol in fermenters [46]. The use of this equipment facilitates the conditions of controlled study by the simultaneous analysis of metabolic processes. The monitoring of $CO_2$ and $O_2$ concentrations is performed in the fermenter continuously and directly, where the fermentation processes take place. The parallel measurement of $CO_2$, $O_2$ and ethanol allows the analysis of metabolic processes without interruption during the entire fermentation process.

2.2.8. Experimental Design and Statistical Analysis

The experiment was conducted into three factor full factorial experiment for each type of carbohydrate (glucose, fructose and xylose) and total of carbohydrates. Each independent variable (concentration of sulphuric acid, ratio S/L and enzymes dosage) had 3 levels, as follows: concentration of sulphuric acid (1%, 2% and 3% respectively), ratio S/L (1/20 *w/v*, 1/25 *w/v* and 1/30 *w/v* respectively) and enzymes dosage (30/30 μL, 60/60 μL and 90/90 μL). The response of design were considered total of carbohydrates, glucose, fructose and xylose. The full factorial designed was made using Design-Expert 10.0 (Stat-Ease, Inc., Minneapolis, MN, USA).

The model used to predict the evolution of carbohydrates was a quadratic polynomial response surface model which can be applied to fit the experimental data obtained by Box-Behnken design. The quadratic polynomial response surface model which describes the relationship between the experimental data is [47,48]:

$$Y = a_o + \sum_{i=1}^{n}(a_i X_i) + \sum_{i=1}^{n}\left(a_{ii} X_{ii}^2\right) + \sum_{i=1}^{n}\left(a_{ii} X_i X_j\right) \tag{1}$$

where: $Y$—predicted response, $X_i$ stands for the coded levels of the design variable (concentration of sulphuric acid, ratio S/L and enzymes dosage—Table 1), $a_o$ is a constant, $a_i$—linear effects, $a_{ii}$—quadratic effects and $a_{ij}$—interaction effects.

**Table 1.** Levels in full factorial experiments for carbohydrates, glucose, xylose and fructose.

| Factors | Level | | |
|---|---|---|---|
| | −1 | 0 | 1 |
| Sulphuric acid (%), $X_1$ | 1 | 2 | 3 |
| Ratio S/L (*w/v*), $X_2$ | 1/20 | 1/25 | 1/30 |
| Enzymes dosage (μL/g of sample), $X_3$ | 30/30 | 60/60 | 90/90 |

The results of acid organics and individual phenolic compounds were presented to analysis of variance (ANOVA) using Statgraphics Centurion XVIII software (Manugistics Corp., Rockville, MD, USA trial version).

## 3. Results and Discussions

### 3.1. Chemichal Composition of WS

Table 2 shows the chemical composition of WS used in the experiments compared to the composition of WS reported in different studies (Table 3).

**Table 2.** The chemical composition of wheat straws (WS) used in the experiments.

| Cellulose (%) | Acid Insoluble Lignin (%) | Acid Soluble Lignin (%) | Dry Substance (%) | Humidity (%) | Ash (%) |
|---|---|---|---|---|---|
| 37.53 ± 1.15 | 14.35 ± 0.53 | 1.42 ± 0.18 | 92.32 ± 0.23 | 7.68 ± 0.54 | 3.87 ± 0.07 |

**Table 3.** The chemical composition of WS reported in different studies.

| Raw Material | Cellulose (%) | Lignin (%) | Ash (%) | Reference |
|---|---|---|---|---|
| | 44.8 ± 0.7 | 8.46 ± 0.31 | 5.68 | [49] |
| | 44.2 ± 1.8 | 22.4 ± 1.7 | 2.8 ± 0.6 | [50] |
| Wheat straws | 39.8 | 22.6 | 4.2 | [51] |
| | 39 | 17 | 1.8 | [52] |
| | 38–40.8 | 8.9–10.5 | 1.4 | [53] |

## 3.2. Scanning Electron Microscope (SEM) Analysis

The scanning electron microscope (SEM) analysis was used to investigate the changes in the structure of WS samples after acid pretreatment. From the images (Figure 1) a significant difference of the structural composition due to pretreatment with $H_2SO_4$ at different concentrations can be observed. The highest degradation of lignocellulosic material was obtained in the case of pretreatment with 3% $H_2SO_4$ (*v/v*) compared to other concentrations of the same acid. Figure 1D shows that initial uniform and rigid structure of the WS was changed after pretreatment, obtaining a porous structure which can positively influence the enzymatic action. This modification has also been reported by Zheng et al. (2018) and Momayez et al. (2019) [24,54].

## 3.3. ATR-FTIR Analysis

The changes in the functional groups as a result of the pretreatment of the straw biomass were analysed by means of attenuated total reflectance-Fourier transform infrared spectroscopy (ATR-FTIR). Figure 2 shows the spectra recorded for the untreated sample, the liquid fraction of the sample following sulfuric acid pretreatment 1%, 2% and 3% (*v/v*) (Figure 2A) and for the solid fractions of the samples after enzymatic hydrolysis (B, C, D). As seen in Figure 2A, the chemical shifts in the liquid fraction of the sample resulted from the sulfuric acid treatment were similar irrespective of the volume of acid used and the S/L ratio and included a high absorbance peak around 3400 cm$^{-1}$ corresponding to O–H stretching, a sharp peak at 1640 cm$^{-1}$ assigned to the C–O bonds in the alkyl groups of lignin side chains [55], and some small peaks around 1200 and 1050 cm$^{-1}$, which were due to C–O stretching vibrations in cellulose and hemicellulose structure [56].

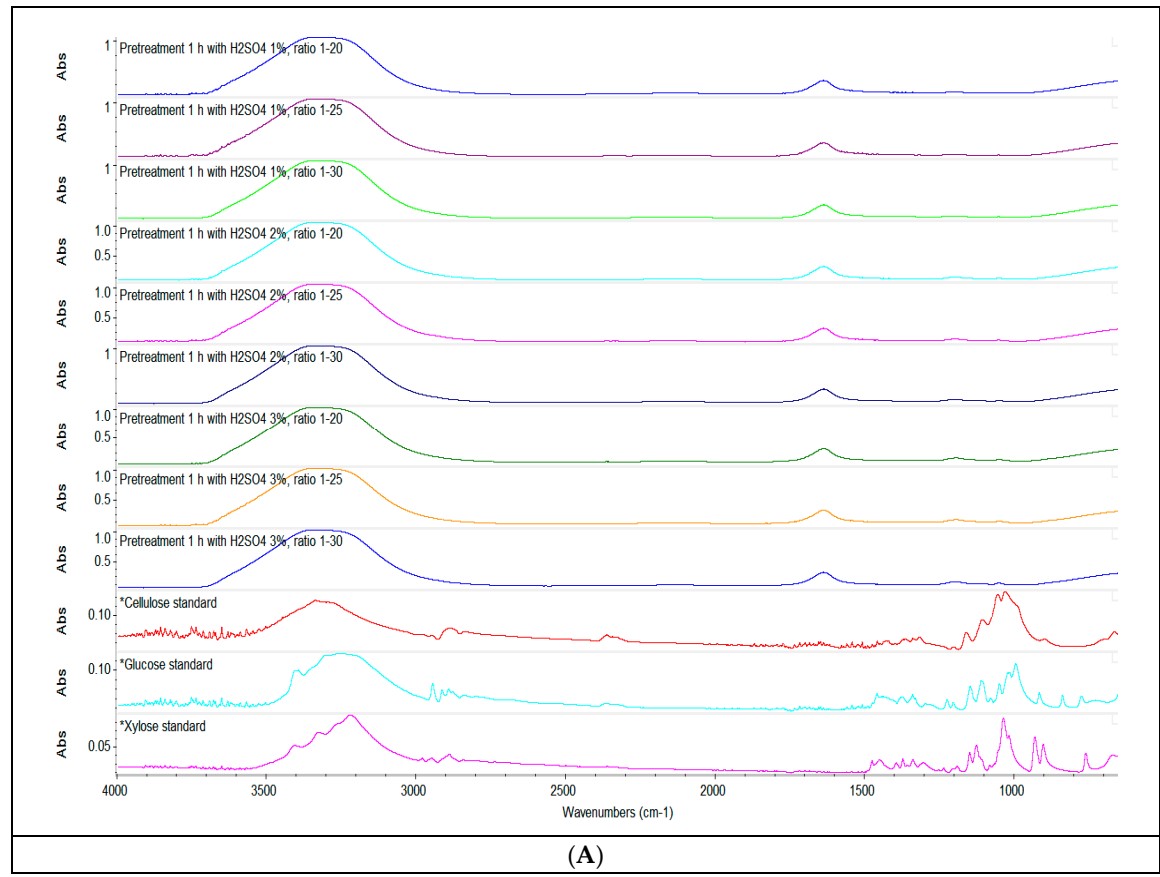

**Figure 2.** *Cont.*

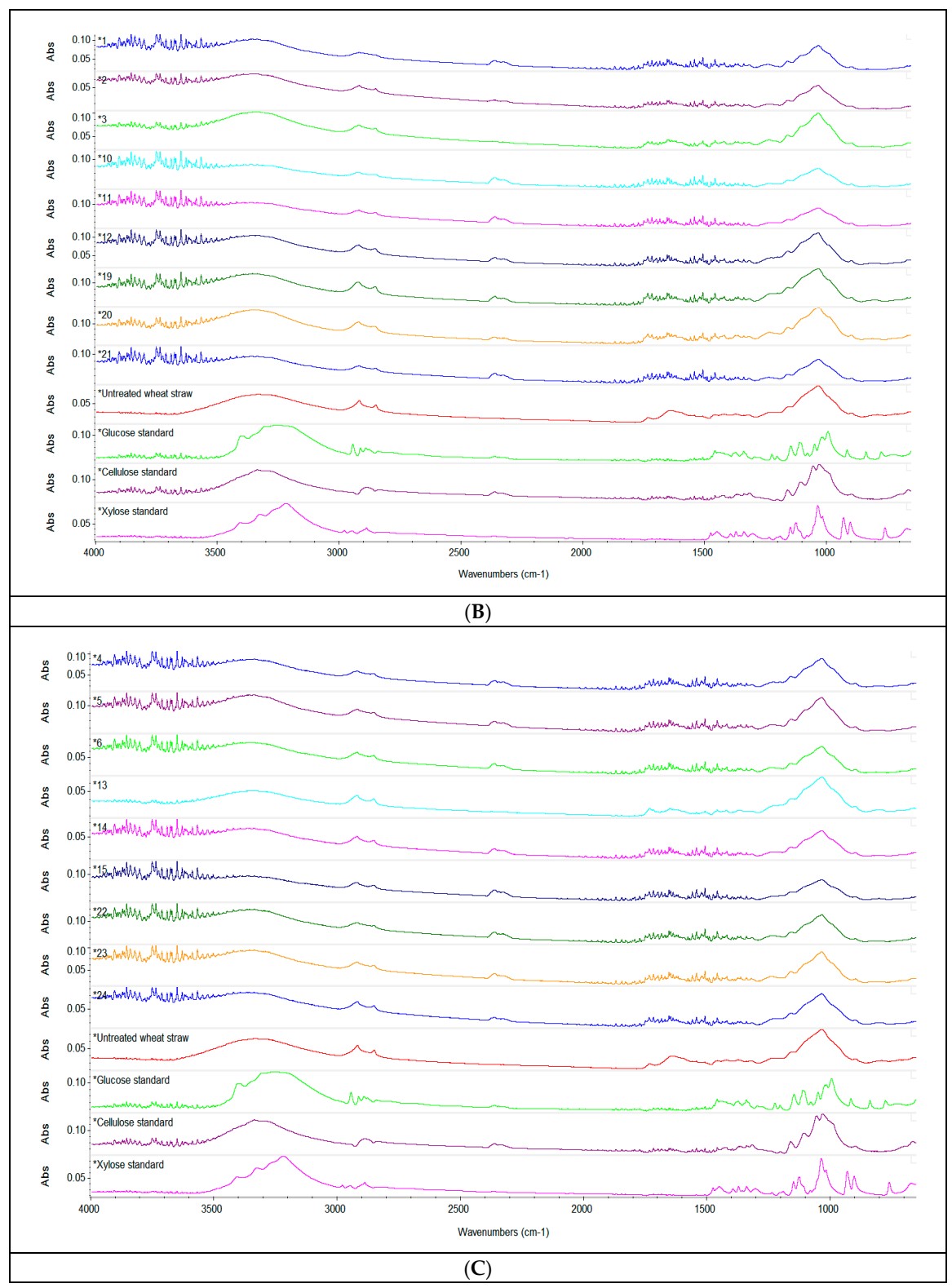

**Figure 2.** *Cont.*

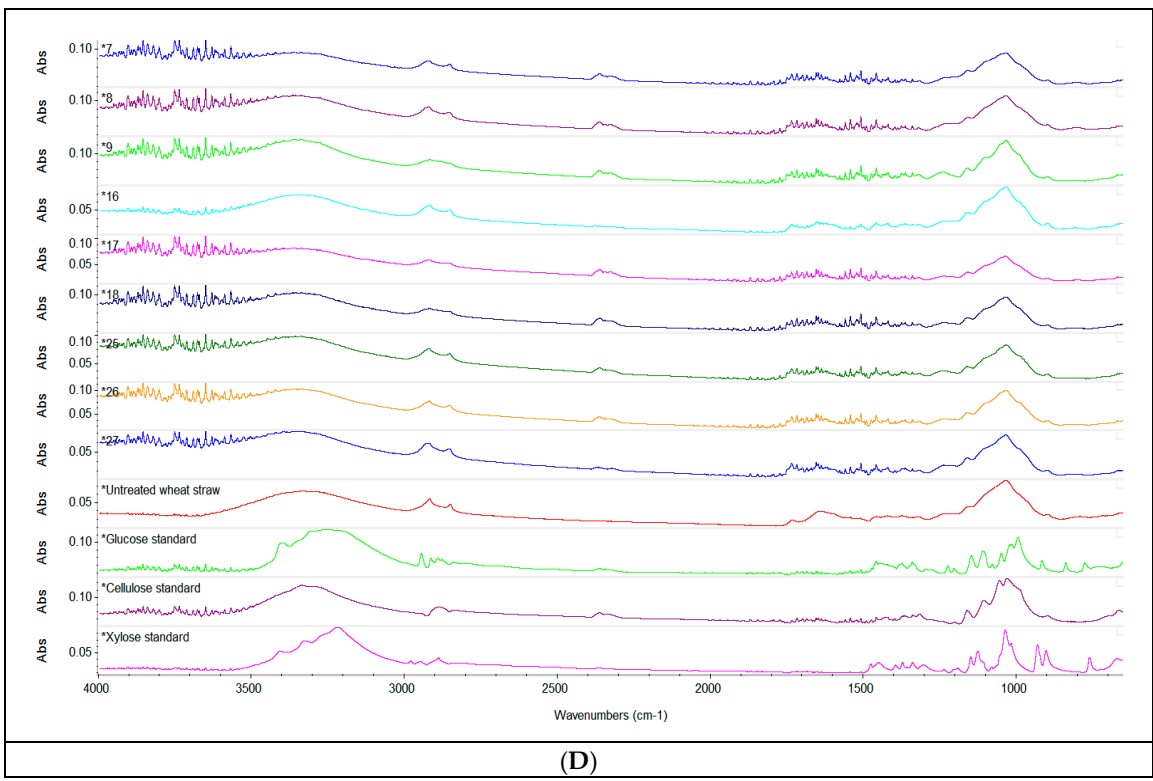

**(D)**

**Figure 2.** Spectroscopic analysis. ATR-FTIR spectra of the liquid fraction after acid pretreatment with 1% H$_2$SO$_4$ (*v/v*), 2% H$_2$SO$_4$ (*v/v*), 3% H$_2$SO$_4$ (*v/v*) (**A**) and of the solid fraction of enzymatically hydrolysed straw (**B**–**D**), compared to the untreated sample and standards of cellulose, xylose and glucose respectively. Depending on the concentration of H$_2$SO$_4$ used, the spectra were stacked to make their interpretation easier.

By comparison to the spectra of the untreated sample, the solid fraction of pretreated and enzymatically hydrolysed straw presented notable changes in the wide absorption band at 3400 cm$^{-1}$ in function of the volume of sulfuric acid, the enzyme dose and the S/L ratio. A decrease of the intensity of the absorption band in this region indicated a reduction of the cellulose content, as previously reported by Zheng et al. (2018) [24]. When considering the conditions of the pretreatment and enzymatic hydrolysis applied, the reduction of cellulose content seems to be more pronounced when an enzyme dose of 90 μL was used. The peaks at 2910 cm$^{-1}$, corresponding to bending vibration of C–H hemicelluloses and cellulose [57], and the peaks around 2300 cm$^{-1}$, also denoting the presence of cellulose in the sample, showed similar changes of intensity determined by the pretreatment and enzymatic hydrolysis. The prominent sharp peak around 1060–1000 cm$^{-1}$ was attributed to C=C, C–O and C–C–O groups stretching in lignin, cellulose and hemicellulose [58]; the highest decrease in the intensity of this peak was also determined by the use of an enzyme dose of 90 μL associated with a higher S/L ratio in the pretreatment and enzymatic hydrolysis.

## 3.4. Determination of Individual Phenolic Compounds

The WS contain a wide range of individual phenolic compounds (caffeic, p-coumaric, 4-hydroxybenzoic, protocatechuic, chlorogenic, vanillic acid etc.,). The phenols are inhibitory compounds for fermentative microorganisms and cellulases in the process of obtaining bioethanol [59]. The polyphenols must be removed before the fermentation to improve the yield of bioethanol in WS and decrease the amount of acid lactic, thus inhibiting the activity of lactic bacteria [60]. During pretreatment, phenolic compounds are removed from the lignocellulosic material because of the action of acid which leads to the breaking of bonds between polysaccharides and polyphenols.

Chen et al. (2018) studied the chemical compounds of rice straws after pretreatment with 2% $H_2SO_4$ (*v/v*) and obtained that p-coumaric acid has the highest concentration (0.40 mg/g) compared to vanillic (0.09 mg/g), chlorogenic (0.02 mg/g) and caffeic acid (0.04 mg/g) [61].

In the studied samples it can be observed that increasing the concentration of sulphuric acid results in an enhancement of the phenolic compounds. Thus, the highest value for the protocatechuic (34.72 ± 0.06 μg/g), vanillic (18.92 ± 0.02 μg/g), 4-hydroxybenzoic (18.74 ± 0.01 μg/g) were obtained at the 3% concentration of sulphuric acid and 1/30 S/L ratio, but p-coumaric (3.21 ± 0.01 μg/g), chlorogenic (3.36 ± 0.04 μg/g) and caffeic acid (3.96 ± 0.03 μg/g) were obtained at the 2% concentration of sulphuric acid and 1/30 S/L ratio. The values for individual phenolic compounds are shown in Table 4 (the chromatograms obtained for standards and one of the samples can be found in Supplementary Materials).

### 3.5. Determination of Organic Acids

The presence and amount of organic acids depend on the nature of the material and the conditions of the solvent pretreatment. The significant parameter that improves the formation of organic acids are the pH with high values and acid pretreatment which imply the risk of production of furaldehydes (FF and HMF) and aliphatic acids (formic and acetic acid) [62]. Lu et al. (2010) founded 3300 mg/L acetic acid, which are 10–15 times higher concentrations in comparison with other acids [63]. Also, Erdei et al. (2010) identified a concentration of 1.7 g/L acetic acid in liquid (prehydrolysate) fractions in steam-pretreated (temp. 190 °C for 10 min) WS slurry [64] compared to that of Linde et al.'s (2008) who obtained 0.04–1.01 g/L acetic acid (temp. 190–210 °C, residence time 2–10 min, sulfuric acid 0.2%) [65]. In the study conducted by Djioleu (2015), the highest concentration for acetic acid and formic acid was 11.04 g/L and 6.08 g/L, respectively [66]. Rajan et al. (2014) pretreated WS at 140 °C with 10 $dm^3/m^3$ $H_2SO_4$ concentration for 30 min and have obtained concentrations of formic acid and acetic acid of 32.37 ± 4.91, 7.98 ± 1.02 g/kg, respectively [67]. In another study Rajan et al. (2014) pretreated rice straws at 220 °C for 52 min, pH 7.0 and have obtained concentrations of formic acid and acetic acid of 6.32 ± 1.46 and 8.45 ± 0.59 g/L, respectively [68].

In this study, organic acids from WS samples pretreated with $H_2SO_4$ of different concentrations (1, 2 and 3% (*v/v*)) were analysed and observed that the highest content of acetic acid (0.94 ± 0.01 mg/g) was founded at the correlation between 3% $H_2SO_4$ (*v/v*) and 1/30 ratio, but the lowest values of acetic acid (0.40 ± 0.02 mg/g) was identified at the pretreatment with 1% $H_2SO_4$ (*v/v*) and 1/30 ratio. The content of organic acids are shown in Table 5 (the chromatograms obtained for standards and one of the samples can be found in Supplementary Materials).

The lowest value of gluconic acid was obtained at 1% (*v/v*) $H_2SO_4$, s/l ratio 1/20—13.77 ± 0.05 mg/g, but highest value at 3% (*v/v*) $H_2SO_4$, s/l ratio 1/30—22.59 ± 0.05 mg/g. Gluconic acid is present in various plants, fruits, wine and honey. The obtained concentration of gluconic acid can be explained by the oxidation of aldehyde group (C1) of D-glucose to a carboxyl group [69,70].

### 3.6. Determination of Individual Carbohydrates After Enzymatic Hydrolysis

Saha and Cotta (2010) analysed the content of neutral monosaccharides in WS after pretreatment with 0.75% $H_2SO_4$ (*v/v*) at 121 °C for 1 h and hydrolysis (using three commercial enzyme preparations Celluclast 1.5, Novozym 188 and ViscoStar 150 L) obtained the following concentration for glucose—282 mg/g, xylose—180 mg/g and total content of carbohydrates—504 mg/g [71]. Saha et al. (2005) were established from barley straw a concentration of 214 mg/g for glucose, 208 mg/g for xylose+galactose and 452 mg/g for total carbohydrates after pretreatment with 1% (*v/v*) $H_2SO_4$ at 121 °C for 1 h and hydrolysis using the same three commercial enzyme preparations [15]. Analysing different varieties of rice straws, Park et al. (2011) showed that these contain free fructose with values between 7.4 ± 0.1–22.9 ± 0.2 g/Kg [72]. Also, Park et al. (2009) obtained 0.62–2.32% fructose (per dry weight of rice straw (*w/w*)) [73].

**Table 4.** Content of phenolic compounds in pretreated WS with different acid concentrations and variations of the S/L ratio (μg/g sample).

| Phenolic Compounds | Pretreatment $H_2SO_4$ 1%, s/l ratio 1/20 | Pretreatment $H_2SO_4$ 1%, s/l ratio 1/25 | Pretreatment $H_2SO_4$ 1%, s/l ratio 1/30 | Pretreatment $H_2SO_4$ 2%, s/l ratio 1/20 | Pretreatment $H_2SO_4$ 2%, s/l ratio 1/25 | Pretreatment $H_2SO_4$ 2%, s/l ratio 1/30 | Pretreatment $H_2SO_4$ 3%, s/l ratio 1/20 | Pretreatment $H_2SO_4$ 3%, s/l ratio 1/25 | Pretreatment $H_2SO_4$ 3%, s/l ratio 1/30 | *F* Value |
|---|---|---|---|---|---|---|---|---|---|---|
| Protocatecuic acid | 24.73 ± 0.07 (0.35) [h] | 25.94 ± 0.03 (0.37) [g] | 28.94 ± 0.04 (0.41) [e] | 27.045 ± 0.05 (0.39) [f] | 30.00 ± 0.08 (0.43) [d] | 32.59 ± 0.09 (0.47) [b] | 28.23 ± 0.05 (0.40) [e] | 31.53 ± 0.02 (0.45) [c] | 34.72 ± 0.06 (0.49) [a] | 118.71 *** |
| 4-hidroxibenzoic acid | 14.51 ± 0.04 (0.21) [f] | 15.11 ± 0.02 (0.22) [e] | 16.91 ± 0.03 (0.24) [c] | 14.9 ± 0.06 (0.21) [e,f] | 15.93 ± 0.01 (0.23) [d] | 16.52 ± 0.04 (0.23) [c] | 15.86 ± 0.02 (0.23) [d] | 18.1 ± 0.01 (0.26) [b] | 18.74 ± 0.01 (0.27) [a] | 75.77 *** |
| Vanillic acid | 9.38 ± 0.05 (0.13) [e] | 9.79 ± 0.05 (0.14) [e] | 9.81 ± 0.04 (0.14) [e] | 11.93 ± 0.05 (0.17) [d] | 12.36 ± 0.04 (0.18) [c] | 14 ± 0.03 (0.20) [b] | 12.71 ± 0.02 (0.18) [c] | 18.53 ± 0.02 (0.26) [a] | 18.92 ± 0.02 (0.27) [a] | 690.71 *** |
| Cafeic Acid | 3.12 ± 0.02 (0.04) [d] | 3.37 ± 0.05 (0.05) [c] | 3.86 ± 0.04 (0.06) [a] | 3.22 ± 0.04 (0.04) [d] | 3.50 ± 0.03 (0.05) [b] | 3.96 ± 0.03 (0.06) [a] | 2.84 ± 0.02 (0.04) [e] | 3.35 ± 0.01 (0.05) [c] | 2.9 ± 0.03 (0.04) [e] | 127.72 *** |
| Clorogenic acid | 2.74 ± 0.01 (0.04) [e] | 3.01 ± 0.03 (0.04) [c] | 3.24 ± 0.02 (0.05) [b] | 2.8 ± 0.02 (0.04) [d,e] | 3.06 ± 0.02 (0.04) [c] | 3.36 ± 0.04 (0.05) [a] | 2.86 ± 0.02 (0.04) [d] | 3.1 ± 0.03 (0.04) [c] | 3.21 ± 0.01 (0.04) [b] | 46.46 *** |
| p-cumaric acid | 2.74 ± 0.01 (0.04) [g] | 3.01 ± 0.01 (0.04) [d,e] | 3.24 ± 0.02 (0.04) [c] | 2.8 ± 0.03 (0.04) [f] | 3.06 ± 0.02 (0.04) [d] | 3.36 ± 0.04 (0.04) [b] | 2.86 ± 0.03 (0.04) [e,f] | 3.1 ± 0.01 (0.04) [a,b] | 3.21 ± 0.01 (0.04) [a] | 105.81 *** |

ns–not significant ($p > 0.05$), * $p < 0.05$, ** $p < 0.01$, *** $p < 0.001$, a–h–different letters in the same row indicate significant differences between samples ($p < 0.001$).

**Table 5.** Content of organic acids in pretreated straws with different acid concentrations and variations of the S/L ratio (mg/g sample).

| Organic Acids | Pretreatment $H_2SO_4$ 1%, s/l ratio 1/20 | Pretreatment $H_2SO_4$ 1%, s/l ratio 1/25 | Pretreatment H2SO4 1%, s/l ratio 1/30 | Pretreatment $H_2SO_4$ 2%, s/l ratio 1/20 | Pretreatment $H_2SO_4$ 2%, s/l ratio 1/25 | Pretreatment $H_2SO_4$ 2%, s/l ratio 1/30 | Pretreatment $H_2SO_4$ 3%, s/l ratio 1/20 | Pretreatment $H_2SO_4$ 3%, s/l ratio 1/25 | Pretreatment $H_2SO_4$ 3%, s/l ratio 1/30 | *F* Value |
|---|---|---|---|---|---|---|---|---|---|---|
| Gluconic acid | 13.77 ± 0.05 (196.84) [g] | 15.80 ± 0.07 (190.13) [e] | 18.49 ± 0.05 (178.45) [c] | 14.80 ± 0.1 (240.11) [f] | 16.35 ± 0.8 (233.64) [d] | 20.23 ± 0.06 (203.34) [b] | 15.71 ± 0.02 (127.66) [e] | 20.50 ± 0.06 (292.95) [b] | 22.59 ± 0.05 (279.97) [a] | 366.72 *** |
| Formic acid | 1.03 ± 0.02 (14.71) [h] | 1.27 ± 0.03 (18.18) [g] | 1.49 ± 0.02 (21.36) [e] | 1.37 ± 0.05 (19.59) [f] | 1.67 ± 0.03 (23.94) [d] | 1.93 ± 0.05 (27.60) [c] | 1.95 ± 0.06 (27.90) [c] | 2.38 ± 0.05 (34.08) [b] | 2.69 ± 0.02 (38.43) [a] | 851.02 *** |
| Acetic acid | 0.41 ± 0.03 (5.94) [g] | 0.41 ± 0.01 (5.88) [g] | 0.40 ± 0.02 (5.79) [g] | 0.48 ± 0.07 (14.52) [f] | 0.71 ± 0.03 (4.80) [d] | 0.81 ± 0.04 (11.65) [c] | 0.65 ± 0.02 (18.38) [e] | 0.87 ± 0.07 (19.71) [b] | 0.94 ± 0.01 (53.57) [a] | 205.44 *** |

ns–not significant ($p > 0.05$), * $p < 0.05$, ** $p < 0.01$, *** $p < 0.001$, a–h–different letters in the same row indicate significant differences between samples ($p < 0.001$). The *F*-value represents that the analysis of variance (ANOVA) is statistically significant. The obtained values of this parameter for content of phenolic compounds and content of organic acids represent the significant values of the test.

In this study, the Box Behnken design based on the response surface methodology with the representation of the experimental values of the independent variables was used to identify the optimal condition in order to obtain the highest carbohydrates content.

Figure 3 shows that the highest glucose content was obtained at the correlation of the following parameters—3% $H_2SO_4$ (*v/v*), 1/30 ratio and 90/90 µL/g of added enzyme content, and the lowest—1% $H_2SO_4$ (*v/v*), 1/20 ratio and 30/30 µL/g of added enzyme content. This tendency is also manifested for the other analysed parameters (fructose content–Figure 4, xylose–Figure 5 and the total carbohydrates content—Figure 6).

The equation for glucose content (mg/g sample) is presented in the Equation (2):

$$\begin{aligned} Glucose &= 59.5633 + 1.7338{\cdot}X_1 + 24.3427{\cdot}X_2 + 2.2761{\cdot}X_3 - 0.1350{\cdot}X_1{\cdot}X_2 \\ &\quad +0.1666{\cdot}X_1{\cdot}X_3 + 0.2800{\cdot}X_2{\cdot}X_3 + 0.2050{\cdot}X_1^2 \\ &\quad -2.1916{\cdot}X_2^2 - 0.7816{\cdot}X_3^2 \end{aligned} \tag{2}$$

where: $Y_1$—the value of the glucose content parameter, mg/g sample; $X_1$, $X_2$, $X_3$—coded values for sulfuric acid content (%), S/L ratio (*w/v*) and added enzyme content (µL/g sample).

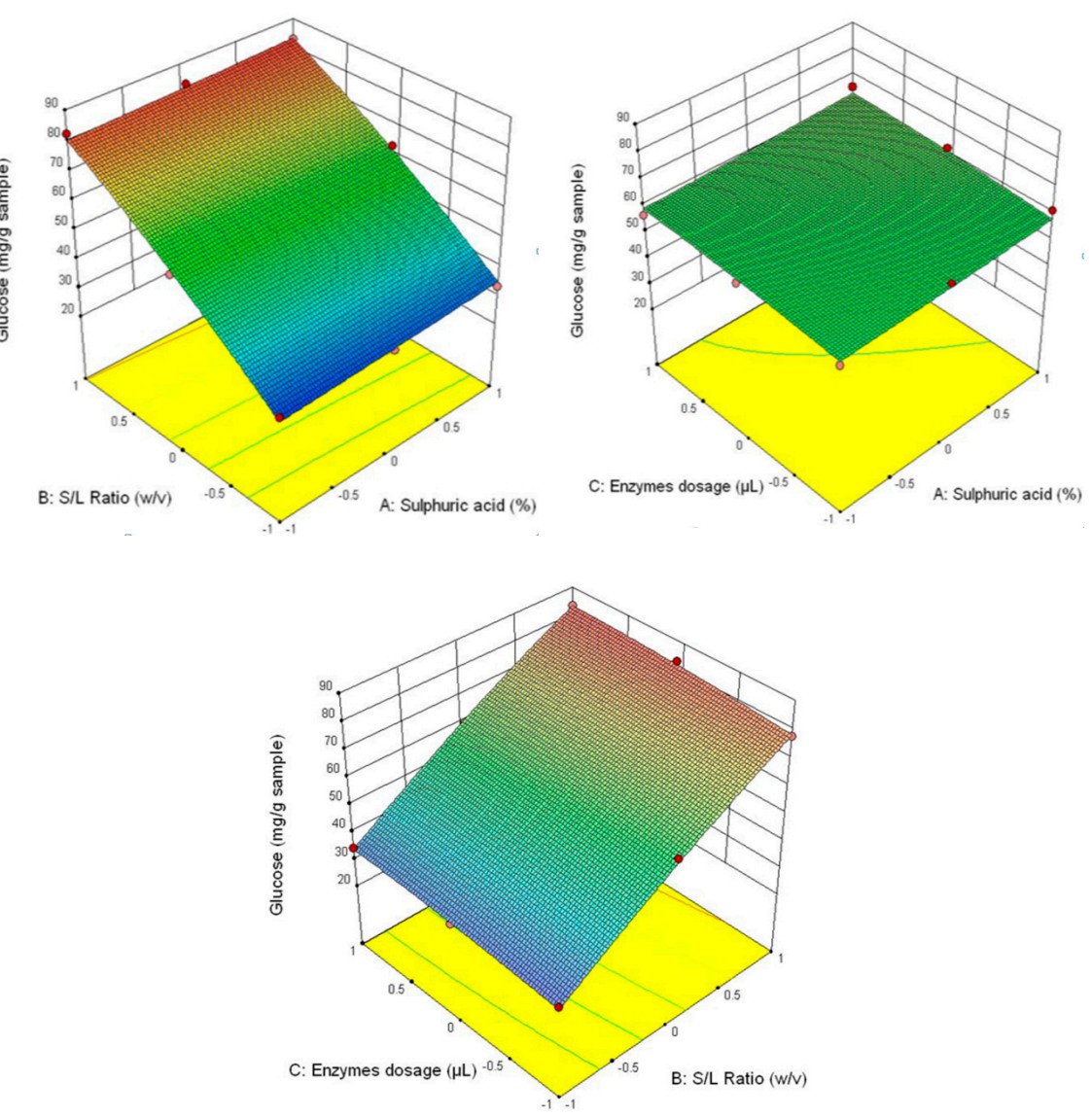

**Figure 3.** 3D diagrams of glucose content.

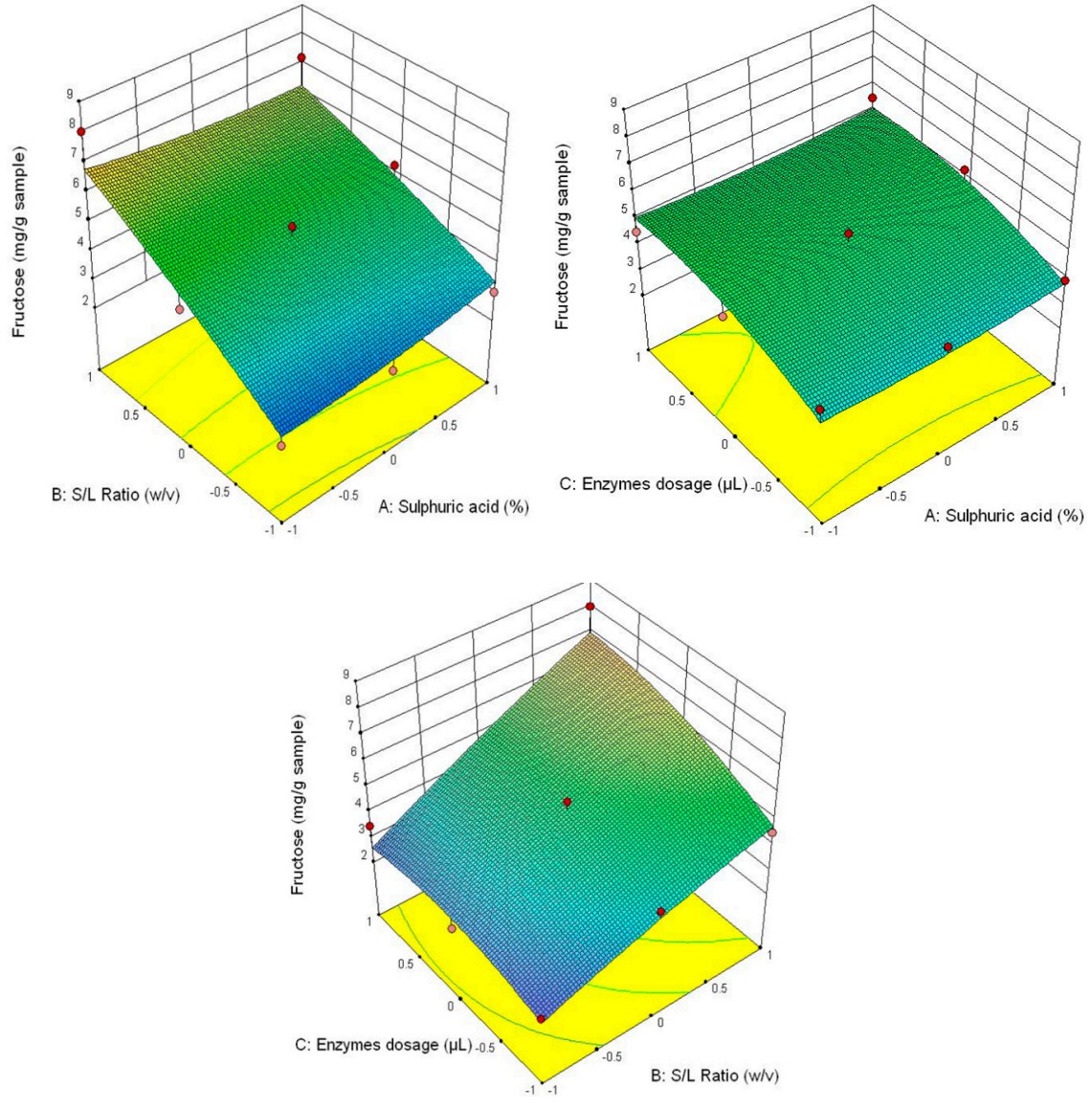

**Figure 4.** 3D diagrams of fructose content.

The equation for fructose content (mg/g sample) is presented in the Equation (3):

$$Fructose = 4.7733 + 0.0277 \cdot X_1 + 1.7327 \cdot X_2 + 0.6527 \cdot X_3 - 0.3141 \cdot X_1 \cdot X_2 + \\ 8.3 \cdot 10^4 \cdot X_1 \cdot X_3 + 0.4766 \cdot X_2 \cdot X_3 + 0.1533 \cdot X_1^2 - 0.1516 \cdot X_2^2 - 0.4916 \cdot X_3^2 \tag{3}$$

where: $Y_1$—the value of the fructose content parameter, mg/g sample; $X_1$, $X_2$, $X_3$—coded values for sulfuric acid content (%), S/L ratio (*w/v*) and added enzyme content (µL/g sample).

The equation for xylose content (mg/g sample) is presented in the Equation (4):

$$Xylose = 6.3244 + 2.0000 \cdot X_1 + 2.2583 \cdot X_2 + 0.2638 \cdot X_3 + 0.3458 \cdot X_1 \cdot X_2 \\ + 0.3175 \cdot X_1 \cdot X_3 + 0.1358 \cdot X_2 \cdot X_3 + 1.3333 \cdot X_1^2 \\ + 0.8450 \cdot X_2^2 - 0.1116 \cdot X_3^2 \tag{4}$$

where: $Y_1$—the value of the xylose content parameter, mg/g sample; $X_1$, $X_2$, $X_3$—coded values for sulfuric acid content (%), S/L ratio (*w/v*) and added enzyme content (µL/g sample).

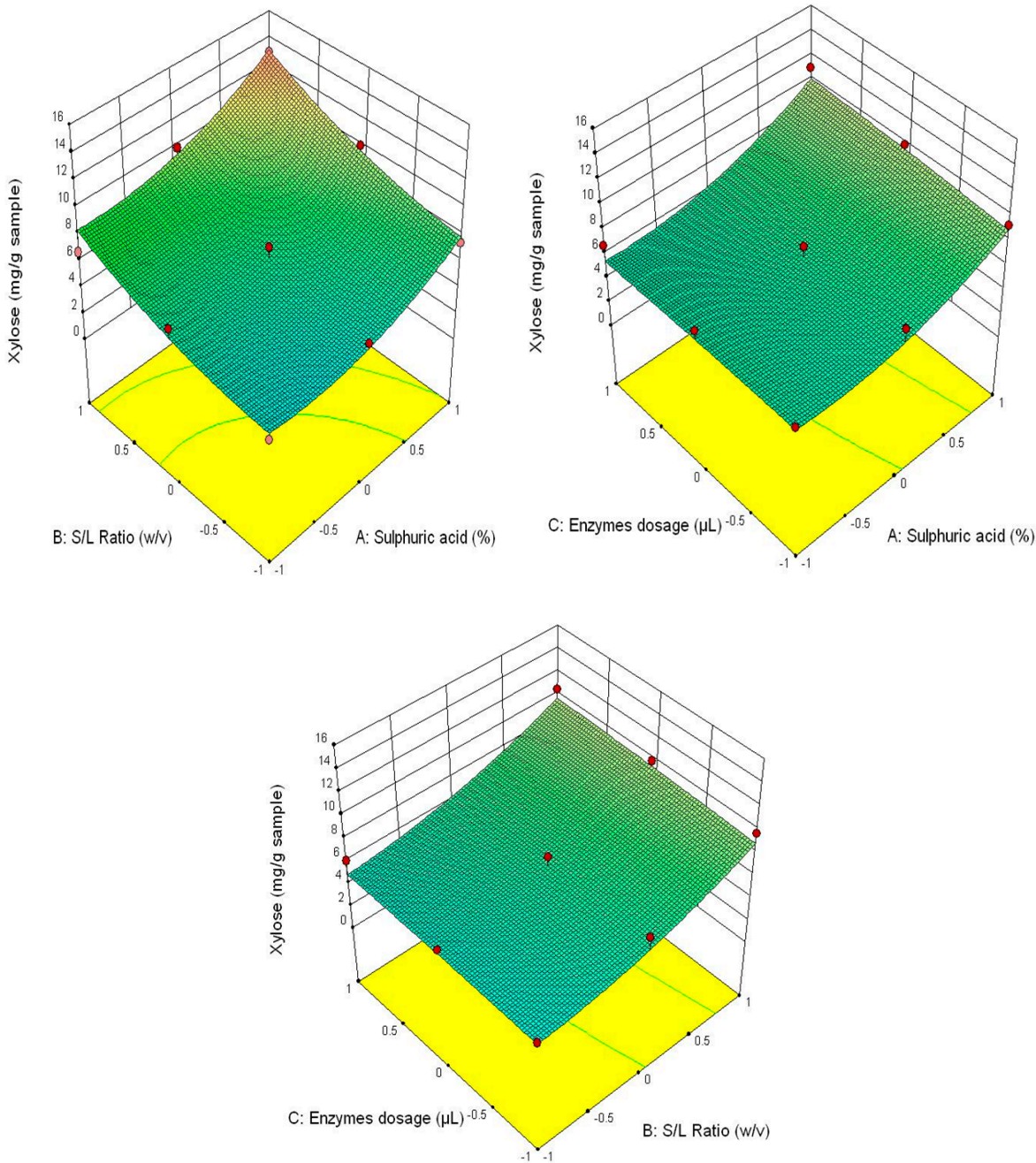

**Figure 5.** 3D diagrams of xylose content.

The equation for carbohydrates content (mg/g sample) is presented in the Equation (5)

$$
\begin{aligned}
Total\ of\ carbohydrates = {}& 70.6592 + 3.7616{\cdot}X_1 + 28.3344{\cdot}X_2 + 3.1911{\cdot}X_3 \\
& -0.1016{\cdot}X_1{\cdot}X_2 + 0.4850{\cdot}X_1{\cdot}X_3 + 0.8916{\cdot}X_2{\cdot}X_3 + 1.6905{\cdot}X_1^2 \\
& -1.4977{\cdot}X_2^2 - 1.3844{\cdot}X_3^2
\end{aligned}
\tag{5}
$$

where: $Y_1$—the value of the carbohydrates content parameter, mg/g sample; $X_1$, $X_2$, $X_3$—coded values for sulfuric acid content (%), S/L ratio (*w/v*) and added enzyme content (μL/g sample).

The statistical parameters of each model are represented in Table 6. All the models proposed are significant ($p < 0.0001$). The coefficents of regression obtained for above quadratic equation indicated

that the variation of glucose, fructose, xylose and total carbohydrates content can be explained by the correlation between independent variables.

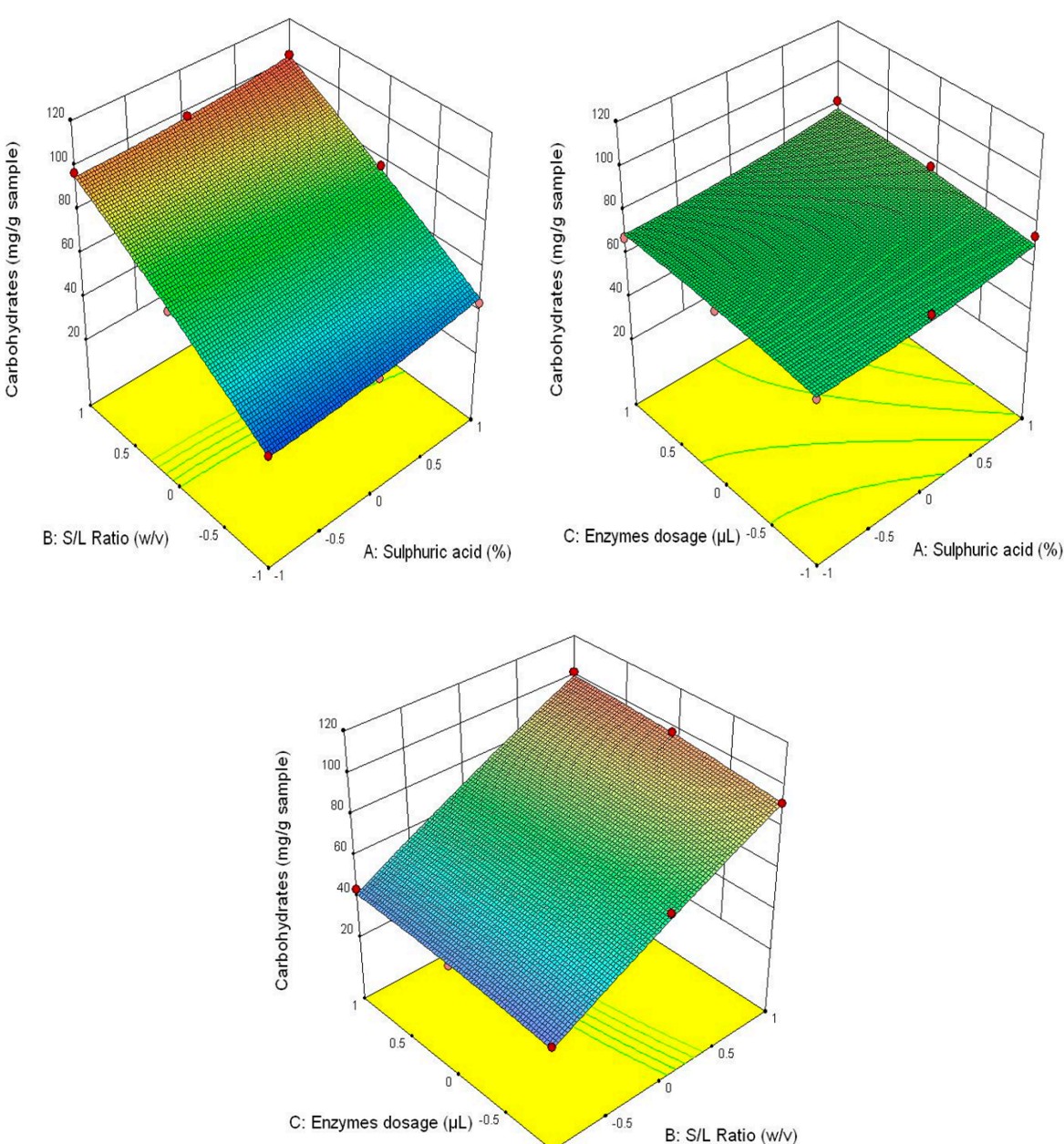

**Figure 6.** 3D diagrams of total carbohydrates content.

**Table 6.** Sum of square, mean square, *F*-value and $R^2$.

| Parameter | Sum of Square | Mean Square | *F*-Value | $R^2$ |
|---|---|---|---|---|
| Carbohydrates | 14,943.73 | 1660.41 | 193.62 | 0.9903 |
| Glucose | 10,847.87 | 1205.32 | 392.16 | 0.9952 |
| Xylose | 182.95 | 20.33 | 7.20 | 0.7922 |
| Fructose | 67.37 | 7.49 | 13.70 | 0.8788 |

$p < 0.0001$.

### 3.7. Monitoring of Bioethanol Concentration

Following the enzymatic hydrolysis, organic acids and phenolic compounds from the total of 7 organic acids and 12 individual phenolic compounds studied were not detected, this is explained by the efficient washing of hydrolysed straw. Bellido et al. (2011) reported that regarding the individual effect of acetic acid, it was observed that the inhibition increased with the concentration of acetic acid and the media containing 3.5 g/L of acetic acid completely inhibited both the yeasts growth and production of ethanol. The theoretical yield of ethanol, defined as the percentage of the total amount of ethanol that could be produced from all available carbohydrates, decreased when acetic acid was present in mash in high concentration [74].

Fermentation Process

Vintilă et al. (2010) using a similar equipment of BlueSens sensors reported that after pretreatment with 2% NaOH of the raw material and applying the SSF process, after 48 h a maximum concentration of 1.5% ethanol was obtained in the case of corn ostriches, while for wheat straw the highest concentration of bioethanol was 1.33%. The enzymes used in the study were Trichoderma cellulases Onozuka (15 units/g cellulose), Aspergillus cellulases (15 units/g cellulose), Aspergillus cellobiase Novo (90 units/g cellulose) and the yeast strain was *Saccharomyces cerevisiae CMIT2*. [75]. Saha and Cotta (2006) obtained 15.1 mg/g ethanol following the fermentation process after enzymatic hydrolysis and alkaline pretreatment combined with $H_2O_2$ [76].

Patel et al. (2019) used a combined treatment (microwave-assisted and 0.5% NaOH), 5 FPU/g commercial cellulase (SIGMA) + 5 FPU/g of in-house enzyme (*Aspergillus niger ADH 11*) and followed by SSF with *Sacchromyces cerevisiae 3570*. After 34.42 h a maximum ethanol concentration and productivity of 32.44 g/L (0.95 g/L/h) was obtained from WS with a yield of 0.30 g ethanol/g reducing sugar consumed [77].

Novy et al. (2015) pretreated the wheat straw by steam explosion at 200 °C, 15 bar for 10 min, with a water to wheat straw ratio of 1, added 30 FPU/g (T. reesei SVG17) and followed by SHCF at 30 °C, pH 4.5 with S. cerevisiae IBB10B05. The ethanol obtained was 71.2 g/kg WS [78].

Singhania et al. (2014) applied a pretreatment with dilute acid (2.5% (*w/w*) $H_2SO_4$) on WS, 20 FPU/g Sacchari-SEB-C6 (advanced enzyme) + 20 FPU/g of in-house cellulase (*Penicillium janthinellum*) and followed by SSF with *K. marxianus* MTCC 4136 at 40 °C. After 48 h of fermentation resulted 12 g/L ethanol [79].

Xu et.al. (2011) pretreated with the ratio of WS to liquid at 80 g/kg, the NaOH concentration of 10 kg/m$^3$, the microwave power of 1000 W for 15 min, prehydrolysis at 50 °C for 24 h with Cellubrix® L, pH = 4.8, followed by SSF with *Sacchromyces cerevisiae* at 32 °C. After 120 h, the ethanol yield was 148.93 g/kg WS compared to the untreated material which was only 26.78 g/kg [80].

In this study, the activity of yeast *Saccharomyces cerevisiae* (DistillaMax SR) was not inhibited by the presence of acetic acid, as it was not identified after enzymatic hydrolysis in the analysed samples.

After inoculation of the slurry with the yeast *Saccharomyces cerevisiae* (DistillaMax SR) it can be seen based on the graph generated by the BACVis software that the fermentation process has started. The maximum fermentation was recorded approximately 12–13 h after the inoculation of the yeast, where the highest consumption of $O_2$ (from 22% vol. initially to approx. 15.1% vol.) and carbohydrates and the releasing of $CO_2$ (approx. 26.3% vol.), and the bioethanol content was approx. 0.853% vol. After 12–13 h from the maximum of the fermentation process, it can be observed that toward the end of the fermentation, the concentration of ethyl alcohol increased by approximately 0.4% vol. The maximum concentration of bioethanol obtained in gaseous phase from the fermentation process under the conditions presented was 1.20% (*v/v*) (Figure 7), which means the yield of bioethanol was 47.61 ± 2.3 g/Kg WS.

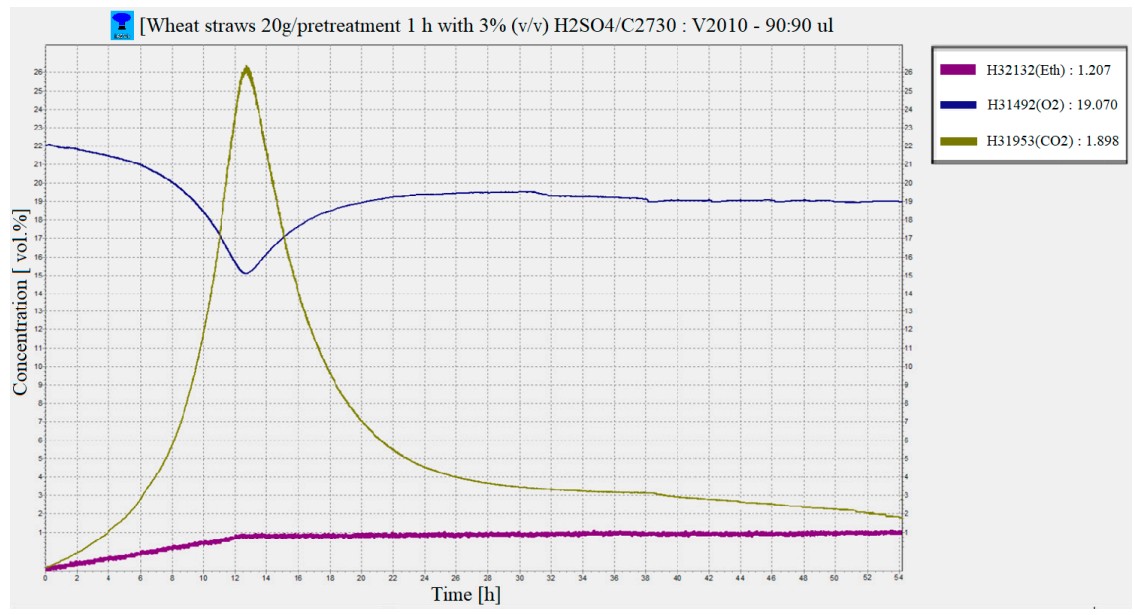

**Figure 7.** Evolution of $CO_2$, $O_2$ and ethanol concentrations during the saccharification and fermentation (SSF) process of WS mash.

## 4. Conclusions

Our results highlighted a highest degradation of lignocellulosic material in the case of pretreatment with 3% (*v/v*) $H_2SO_4$ compared to other concentrations of the same acid. This pretreatment led to an increasing of phenolics and organic acids content, especially of protocatechuic ($34.72 \pm 0.06$ µg/g), vanillic ($18.92 \pm 0.02$ µg/g) and 4-hydroxybenzoic ($18.74 \pm 0.01$ µg/g), while the value of acid gluconic was $22.59 \pm 0.05$ mg/g, formic acid was $2.69 \pm 0.02$ mg/g and acetic acid was $0.94 \pm 0.01$ mg/g. The SEM microstructure of the pretreated samples revealed the changes which occurred on the surface of the straws undergoing pretreatment. Significant results were obtained in the case of treatment with 3% (*v/v*) $H_2SO_4$. ATR-FTIR analysis showed reduction in the intensity of the peaks characteristic to cellulose, hemicellulose and lignin at increased enzyme doses and S/L ratios, thus confirming the efficiency of the pretreatment and the enzymatic hydrolysis. This study showed that the pretreatment with 3% (*v/v*) $H_2SO_4$, in a S/L ratio of 1/30 (*w/v*) and with a dose of enzyme of 90/90 µL/g wheat straw led to the highest concentrations of carbohydrates (xylose $14.31 \pm 0.11$ mg/g, fructose $5.94 \pm 0.13$ mg/g, glucose $84.75 \pm 0.23$ mg/g and total carbohydrates 105.01 mg/g) and as a result the highest ethanol yield after the fermentation process.

The yield of ethanol ($47.61 \pm 2.3$ g/Kg WS) could be improved if the concentration of $H_2SO_4$ (*v/v*) is increased in the pretreatment step, which would probably lead to a more significant disruption of the complex structure of the WS, hence facilitating the action of enzyme mixtures.

**Supplementary Materials:** The following are available online at http://www.mdpi.com/2076-3417/10/21/7638/s1, Figure S1. HPLC—DAD chromatogram at 280 nm (A) and 320 nm (B) for standard (100 mg/l) for gallic acid—peak 1 (8.69 min), protocatechuic acid—peak 2 (15.953 min), 4-hydroxybenzoic acid—peak 3 (20.886 min), vanillic acid—peak 5 (25.631 min), caffeic acid—peak 4 (23.277 min), chlorogenic acid—peak 6 (25.831 min), p-coumaric acid—peak 7 (32.011 min), rosmarinic acid—peak 8 (39.69 min), myricetin—peak 9 (43.216 min), luteolin—peak 10 (49.737 min), quercetin—peak 11 (50.128 min) and kaempferol—peak 12 (56.52 min) and (C—D) content of individual phenolic compounds in acid-pretreated wheat straw (liquid fraction)., Figure S2. HPLC-RID chromatogram for standard carbohydrates mix (CAR10-KIT) (A) and hydrolysed WS, D − (−) − Ribose—peak 1 (3.738 min), D − (+) − Xylose—peak 2 (4.235 min), D − (−) − Arabinose—peak 3 (4.771 min), D − (−) − Fructose—peak 4 (5.298 min), D − (+) − Mannose—peak 5 (5.921 min), D − (+) − Glucose—peak 6 (6.273 min), D − (+) − Galactose—peak 7 (6.77 min), Sucrose—peak 8 (9.886 min), D − (+) − Maltose—peak 9 (12.155 min), α-Lactose monohydrate—peak 10 (14.302 min) and (B) the carbohydrates content of enzymatically hydrolysed WS, Table S1. Box-Behnken design with coded values.

**Author Contributions:** Conceptualization, V.-F.U.; formal analysis, V.-F.U.; funding acquisition, G.G.; investigation, V.-F.U.; methodology, V.-F.U. and G.G.; project administration, G.G.; software, V.-F.U.; supervision, G.G.; validation, G.G.; writing—original draft, V.-F.U.; writing—review and editing, V.-F.U. and G.G. All authors have read and agreed to the published version of the manuscript.

**Funding:** This work was supported by contract no. 18PFE/16.10.2018 funded by Ministry of Research and Innovation within Program 1: Development of national research and development system, Subprogram 1.2: Institutional Performance—RDI excellence funding projects.

**Conflicts of Interest:** The authors declare no conflict of interest.

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
