# Peer review of "Production of Cellulosic Ethanol from Enzymatically Hydrolysed Wheat Straws"

_applsci, doi:10.3390/app10217638_

Round 1

Reviewer 1 Report

Dear authors,

I reviewed your manuscript entitled „Bioethanol production from enzymatically hydrolysed wheat straws”. First of all, I would like to express my acknowledgements for the experimental work you did. I know that obtaining those data took a lot of effort and time!

However, in my view, the work can not be published in the present form because of the following major reasons:

  • The whole manuscript lacks in emphasizing what is the new aspect of the research. Even a lot of other literature is cited, it has not been worked out, which additional value this work brings to the scientific community.
  • The article is hard to read. A lot of rearrangements in the text have to be performed to bring it on a high scientific level. A lot of passages and figures are of minor importance; however, some other important parts have to be added.

I have following detailed comments:

L2: The title is in my view to general. There have been numerous other works dealing with bioethanol from wheat straw.

L13: “Romanian wheat straw” Is there any difference in chemical structure between “Romanian” and other wheat straw that makes it necessary to emphasize that it is Romanian?

L14: “sulphuric acid (1, 2 and 3%)” Is the value in wt.% or in vol.%?

L26: “1.207 %vol” I am sure that this number of digits is not covert by the measuring inaccuracy of the equipment. 3 digits are sufficient.

L12-26: Please add in the abstract, what of your work is original/new for the scientific community. E.g. could the ethanol yield be improved in comparison to other studies?

L30: The current introduction text should be shorted significantly and more relevant info should be added. It must be worked out, what is the missing knowledge in the literature, and how the article contributes in increasing knowledge.

L60-85, 90-98: I don’t see a close connection of these info to the research of the article. It can be removed.

L53: “worldwide cereal production is 2789.8 mln tons. The worldwide cereal production in 2020 (2708.5 mln tons)” For the same fact two different numbers are given. What number is correct?

L59, 90: Fig 1 and 2 are not essential for the article and can be removed.

L99: The given info is a repetition of L87.

L99-113: general statements about pretreatment which are state of the art. There is no need to tell this the reader, as this is a research article and no review.

L135: A figure of ground biomass is not essential for the article, in my view, and can be removed.

L136: Please add, in which device the pretreatment was performed. Was mixing applied? Were the ground samples from pretreatment A used?

L138: “Then, WS were boiled for 1 h at 100 ºC in a water bath” I consider that the 100°C were the pretreatment temperature, correct? Please write that clearer. Was an active cooling applied afterwards?

L141: „filtered under vacuum” may name the filter material and size.

 L145: In which device the hydrolysis was performed?

L146: Were the liquid and solid separated from each other after hydrolysis?

L148,152: Were the samples dried before analysis?

L164: “by Pauliuc et al. (2020) with some modifications [25,26,27].” No named reference number matches to Pauliuc et al.

L173: “1 mL of solution pretreated with acid samples with 1 mL of solution enzyme-hydrolyzed samples for 24 h” The second “with” is surely a mistake.

L176: “(Shimadzu, 176 Kyoto, Japan) with a diode array detector (DAD). The separation of organic acids was performed  into a Kinetex® 2,6 μm Biphenyl 100 Å column, LC Column 150 x 4.6 mm.” Same device is mentioned above. Can be shortened.

L182: “Determination of individual carbohydrates after enzymatic hydrolysis” The determination of sugars after pretreatment in the liquid phase would be also interesting to the readers. Have you data about that?

L195: “The inactivation of enzymes was achieved by exposing the samples for 5 min at 121 ºC [31].” This was done before analysis, correct? Then shift it to the beginning of the paragraph, please.

L198-207: This part is hard to read because it is partly a repetition of pretreatment and hydrolysis description. May shift these sentences to the pretreatment and hydrolysis section.

L216: Is the ethanol sensor in the liquid or the gas phase? If it is in the gas phase, how the ethanol in liquid phase can be determined?

L221: is the sugar yield or concentration the desired variable?

L241-244: This text better fits to materials and methods section.

L247: Is the data in the table the mean value of multiple determination? If yes, please show the standard derivation.

L249: The temperature profile can be described in 1 sentence in the materials and methods section. No need to show this in a figure.

L253-286: I see no own results in this section. The literature overview should be given in introduction.

How did pretreatment A or B affect the following steps? Was a sample, which not underwent the pretreatment step, subjected to hydrolysis step? Then the effect of pretreatment can be evaluated better.

L275-281: The text gives very general info about pretreatment. As this is no review article, I see no need to tell this the reader.

L298: Which SEM-detector was used for the pictures E,F,G?        

L325: The Fig 6 must be improved. Data is much too small displayed. Nobody can read anything in a printout, because the axis and legends are much too small. Also, too much data is shown and a selection has to be made. Otherwise the text in L302-324 is not understandable.

L331: “WS contain a wide range of individual phenolic compounds” Are these phenols in the WS part of lignin ore are they are directly bond to carbohydrates?

L383: “compounds in acid-pretreated wheat straw” Which pretreatment conditions were applied?

L398: “24.73 (035)h” I don’t understand what the “(035)” and “h” stands for. Please explain that. Also, the F-value should be explained briefly.

L405-411: This info better belongs to the introduction.

L413: „with 0.75% H2SO4” When discussing pretreatment results from the literature, please also mention the temperature and residence time of pretreatment.

L417: I would recommend to show a table with all measured sugar yields of the different experiments.

L418: I see no special need to show Fig 8. Analytics of sugars via HPLC are state of the art. Maybe display this in supporting material.

L419: Fructose is in the samples after hydrolysis. What do the authors think where is comes from? From free fructose in wheat straw or from glucose? This should be discussed with literature.

L437: Table 5 is not interesting for most of the readers. Maybe display this in supporting material.

L461: “The equation for xylose content” Should be “carbohydrate content”.

L483-489: The text gives very general info about hydrolysis. As this is no review article, I see no need to tell this the reader.

L490-507: This paragraph should be shortened and shown in materials and methods section.

L517-520: The text gives very general info about fermentation. As this is no review article, I see no need to tell this the reader.

L521-528: This paragraph should be shortened and shown in materials and methods section.

L529-540: I don’t see a connection to the own results.

L552: Figure 13 legend and axis description should be bigger.

L563: „1.207 vol.%“  Is this the ethanol concentration in gas phase? How much is then the concentration in liquid? Please calculate the ethanol yield. This is an essential information.

L567-578: The conclusion is more written like an abstract.

Author Response

Dear Reviewer,

Thank you for your comments and suggestions for improving the manuscript. We attached the responses file.

Reviewer 2 Report

Lines 71-90 To much information about harvesting of cereal.

Lines 99-100 The same information are presented in lines 87-88.

Line 107 Lack of information about enzymatic hydrolysis.

Line 113 What is the novelty of this work? What was done by other authors in this field?

Line 137 Why was used such a concentration of H2SO4?

Line 145 Why was used such a concentration of enzymes?

Figure 4 In my opinion this figure should be removed from the article.

Figure 5B, C, D Why figures 5B, C, D were presented?

Figure 6. These figures are illegible.

Lines 391-395 Could you give the explanation of your results?

Lines 414-415 More information received by other researches should be added.

Figure 8 From my point of view these figures should be removed.

Lines 435-437 What is the difference between factor (table 4) and code (table 5)?

Lines 439-440 Has anyone proposed earlier similar equations in the literature?

Figures 9 – 12 are illegible.

Lines 494-546 Name of fungi, yeast, bacteria should be written italic.

Figure 13 is illegible.

Author Response

(The authors gave the same response as above.)

Reviewer 3 Report

Dear Authors, I have gone through the Article and I would like to stress a few issues that require to be addressed.

The paper is well organized, however some parts must be improved in order to highlight the novelty of the topic.

The abstract section need to discuss the efficiency and stress the efforts to provide the statistical analysis. When mentioning enzymes, the doses are less significant, while the activity is not mentioned. Please refer.

The introduction section provides a well background to the field, however, the mentioned issues do not cover the novel solutions, the reference list needs to be updated or possibly expanded. Figure 1 is adopted from the literature, why reprinting it and not just cite? This is not an essence of few publications nor your original contribution.

Figure 2 is also adopted, it is not a contribution to the field, either make efforts to cover your original concept, or just cite the graphics.

In the materials section provide information about the enzyme activity and storage, which may be crucial.

In the methods section provide information about the preparation and storage of wheat in order to discourage mold formation.

Provide information about pH of the enzyme cocktail. How was it chosen?

For SEM analysis, was there any sputtering? How were the enzymes removed before FTMIR? residual enzymes may affect the results. For the phenolic compounds, please provide information about the background analysis. The purity of substances must be provided.

The results are presented without SD or other statistical parameters, control is missing.

The resolution of figure 6 needs to be improved. Values in crucial wave numbers should be provided with SD, the information about background should be provided.

For HPLC you need to provide LOD and LOQ, without these parameters, the interpretation is extremely difficult. Some of the compounds may undergo secondary reactions. Please explain why was p-value set at presented values.

Some of the diagrams for BB design, show that the range of experiment was not clearly defined, as extreme values are found at the borders of the range. Why create a BB model when not even SD is provided for a singular measurement?

Please address the given issues and maybe expand the discussion in the conclusion section after expanding the introduction for a better background.

Author Response

(The authors gave the same response as above.)

Reviewer 4 Report

Dear Authors, I have gone through the Article and I would like to stress a few issues that require to be addressed.

The paper is well organized, however some parts must be improved in order to highlight the novelty of the topic.

The abstract section need to discuss the efficiency and stress the efforts to provide the statistical analysis. When mentioning enzymes, the doses are less significant, while the activity is not mentioned. Please refer.

The introduction section provides a well background to the field, however, the mentioned issues do not cover the novel solutions, the reference list needs to be updated or possibly expanded. Figure 1 is adopted from the literature, why reprinting it and not just cite? This is not an essence of few publications nor your original contribution.

Figure 2 is also adopted, it is not a contribution to the field, either make efforts to cover your original concept, or just cite the graphics.

In the materials section provide information about the enzyme activity and storage, which may be crucial.

In the methods section provide information about the preparation and storage of wheat in order to discourage mold formation.

Provide information about pH of the enzyme cocktail. How was it chosen?

For SEM analysis, was there any sputtering? How were the enzymes removed before FTMIR? residual enzymes may affect the results. For the phenolic compounds, please provide information about the background analysis. The purity of substances must be provided.

The results are presented without SD or other statistical parameters, control is missing.

The resolution of figure 6 needs to be improved. Values in crucial wave numbers should be provided with SD, the information about background should be provided.

For HPLC you need to provide LOD and LOQ, without these parameters, the interpretation is extremely difficult. Some of the compounds may undergo secondary reactions. Please explain why was p-value set at presented values.

Some of the diagrams for BB design, show that the range of experiment was not clearly defined, as extreme values are found at the borders of the range. Why create a BB model when not even SD is provided for a singular measurement?

Please address the given issues and maybe expand the discussion in the conclusion section after expanding the introduction for a better background.

Author Response

(The authors gave the same response as above.)

Reviewer 5 Report

In-depth and detailed experiments on bioethanol production from wheat have been conducted. However, we think the following aspects need to be modified

  • The figures and layout in general need to be revised. There are several items that appear to be outputting the results of analysis as they are, making it impossible to read the figures(Figure.6, Figure.9,10,11,12,13).
  • Please consider the degree of validity of the results obtained. There are many previous studies on bioethanol, so please try to compare them.

Author Response

(The authors gave the same response as above.)

Round 2

Reviewer 1 Report

Dear authors,

I reviewed again your manuscript.

In my view, the work still cannot be published in the present form because of the following major reasons:

  • Even if some changes were performed, the manuscript still lacks in emphasizing what is the new aspect of the research. It has not been worked out clearly enough, which additional value this work brings to the scientific community.
  • The article is partly hard to read and to understand. In some chapters the "red line" is missing. The quality of the English has to be improved. Sometimes it is hard to understand what is the meaning of the sentences.

Some details I would also like to mention:

 L24: "to evaluate the potential of using wheat straws as a raw material for production of celullosic ethanol in Romania." I don’t find any text passage which deals with this statement in the discussion/conclusion section.

L111-138: In my view, these paragraphs cite some facts from literature which are not connected well by a “red line”.

L168: “a diluted concentration of 1.3% (v/v) H2SO4 was used” I have not read anything in the MM section about 1.3% H2SO4 concentration.

L354: Please state in MM section, how much measurement repetitions were performed to obtain the data presented in Table 1?

Figure 2: A selection of presented data has to be done. To show >40 spectra is not feasible. Also, the description is not clear, as just numbers are used to label the curves in Fig B-C.

L703: “which means the yield of bioetanol was 47.61±2.3 g/Kg WS” It is not explained how the yield is calculated. Please explain this in MM section. So, the reader can understand how you calculate the ethanol yield from the measured gas concentration of ethanol.

From first review:

“Point 31: How did pretreatment A or B affect the following steps? Was a sample, which not underwent the pretreatment step, subjected to hydrolysis step? Then the effect of pretreatment can be evaluated better. Response 31: No untreated sample was subjected to hydrolysis step.” It would be an improvement for the manuscript if data for hydrolysis of untreated sample would be available.

“Point 40: L417: I would recommend to show a table with all measured sugar yields of the different experiments. Response 40: We added below.” I would recommend to show these data either in the manuscript or at least in the supporting material.

Reviewer 2 Report

I'm satisfied with the authors corrections

Reviewer 3 Report

Thank you for addressing the given comments.

Reviewer 4 Report

Thank you for addressing the given comments.

Reviewer 5 Report

Corrections confirmed.